# Experimental Study on Poisson's Ratio of Silty-Fine Sand with Saturation

Kai Yan [1], Yong Wang [1,*], Xianghua Lai [2], Yanli Wang [3] and Zhiyong Yang [4]

1. State Key Laboratory of Geomechanics and Geotechnical Engineering, Institute of Rock and Soil Mechanics, Chinese Academy of Sciences, Wuhan 430071, China
2. Second Institute of Oceanography, MNR, Hangzhou 310012, China
3. Key Laboratory of Geotechnical Mechanics and Engineering of the Ministry of Water Resources, Yangtze River Scientific Research Institute, Wuhan 430010, China
4. Academy of Railway Sciences Engineering Consult Co., Ltd., Beijing 100081, China
* Correspondence: wangyong@whrsm.ac.cn

**Abstract:** The influence of saturation on the Poisson's ratio $v$ of reservoir sediments has an engineering significance in the field of oil and nature gas exploration. Based on a self-developed combined (BE-EE-RC) test system, under the dehydration path, the Poisson's ratio variation of reservoir silty-fine sand in Hangzhou Bay, China, was investigated. Results show that the P- and S-wave velocities vary non-monotonically with decreasing saturation at different net stresses, and reach a maximum at the optimum saturation $S_{r(opt)}$; Biot's theory with respect to variation in $V_p$ with $S_r$ matches well with the measured e data. With a small amount of gas intrusion, Poisson's ratio of saturated sand shows a sudden drop and gradually stabilizes; then, it attenuates slowly and reaches the minimum value at $S_{r(opt)}$. Once the saturation degree decreases to the level lower than $S_{r(opt)}$, it rapidly increases. Based on the soil–water characteristic curve (SWCC) and mesoscopic evolution of internal pore water morphology, the variation in Poisson's ratio $v$ can be divided into four segments of saturation: the boundary effect stage, the primary transition stage, the secondary transition stage, and the unsaturated residual stage. Ultimately, a prediction model for Poisson ratio's $v$ of the silty-fine sand was proposed to consider the saturation variation.

**Keywords:** Poisson's ratio; saturation; sand; bending element; S-wave; P-wave

## 1. Introduction

Shallow gas buried in marine seabed is an important risk factor causing disasters such as overflow, blowout, fires, and even the overturning of drilling platforms, which seriously threaten the safety of workers [1–3]. The most effective measure to prevent accidents is to accurately identify the gas-bearing strata in advance. Therefore, gas saturation is an important parameter to be determined in the field of oil and nature gas exploration. Based on the difference in Poisson's ratio between different lithologies and different pore fluid media, AVO (Amplitude Versus Offset) technology is widely used in reservoir oil and gas identification as a seismic exploration method [4,5]. Among the important parameters of AVO technology [6], compared to wave velocity, wave impedance, and density, Poisson's ratio has a close correlation with saturation [7–9], thus it is more sensitive to identifying the lithology and fluid of reservoirs, being regarded as a good index to reflect the gas-charged properties [10]. Therefore, to investigate the variation of Poisson's ratio with saturation of reservoir sediments is conducive to inferring the free gas-bearing state in the reservoir.

Poisson's ratio is a fundamental elastic parameter characterizing the mechanical properties of materials, which is defined as the ratio of radial strain $\varepsilon_r$ to axial strain $\varepsilon_a$, as shown in Equation (1). Over the last few decades, the measurement of Poisson's ratio of rock and soil has been studied intensively. Generally, there are two methods to determine Poisson's ratio. The first one is the uniaxial loading test. The radial strain and axial strain are

measured by static loading, and Poisson's ratio is calculated according to the Equation (1), which is described as the static Poisson's ratio. This method is commonly used in geotechnical engineering [11]. The second method is to obtain the dynamic Poisson's ratio by indirectly measuring the P- and S-wave velocities ($V_P$ and $V_S$), and calculating it based on the assumption of classical elastic dynamics, as shown in Equation (2) [12,13]. The former can give the variation of Poisson's ratio with strain over the whole strain range, while the latter can only determine the elastic Poisson's ratio in the elastic strain range.

$$v = -(\varepsilon_r / \varepsilon_a) \tag{1}$$

$$v = \frac{0.5(V_P/V_S)^2 - 1}{(V_P/V_S)^2 - 1} \tag{2}$$

Because it is difficult to obtain the undisturbed sample in reservoirs especially for uncemented reservoir sediments, the dynamic Poisson's ratio is more easily obtained by the field wave velocity test, which is a non-destructive method. Therefore, dynamic Poisson's ratio is usually used in the oil and nature gas exploration to infer the gas saturation of a reservoir [14–16]. Because Poisson's ratio is related to the soil type, density, envelope pressure, porosity, and saturation, the wave velocity is also affected by these factors. Among them, the dependence of Poisson's ratio and wave velocity on saturation has been investigated by several experimental studies [17–19]. Nakagawa et al. [20] reported the dramatic change in Poisson's ratio between dry and saturated sand. Kumar and Madhusudhan [21] measured both the P- and S-wave velocities of sand across the full range of saturation degree, and the results revealed that Poisson's ratio reaches the minimum at the 'optimum degree of saturation'. (Which is defined for sand, corresponding to the maximum value of the shear modulus [22]). Pereira and Fredlund [23] studied the variation in Poisson's ratio of a collapsing soil under unsaturated conditions, and they found that Poisson's ratio increases with increasing saturation and reaches the maximum at the saturated state. Inci et al. [24] used a fast and simple ultrasound method to measure P- and S-wave velocities of three types of compacted clay, and the results revealed that Poisson's ratio increases with increasing saturation, and its variation is lower at high saturation compared to that at low. Patel et al. [25] investigated the effect of water content on Poisson's ratio in single-, double-, and triple-layered soils by measuring P- and S-wave velocities at different envelope pressures using bending element (BE) tests, and they found that Poisson's ratio in fine-grained soils is related to saturation, increasing with saturation, but with less variation at high saturation compared to that at low. Oh et al. [26,27] investigated the variation in Poisson's ratio with saturation using BE tests, and they found that Poisson's ratio is a function of saturation, and its characteristic behavior or relationship is closely related to the soil–water characteristic curve (SWCC). Dong [28] studied Poisson's ratio of silty soil under unsaturated conditions using BE tests, and the results showed that Poisson's ratio decreases as the pore water saturation decreases, and the variation in Poisson's ratio with water content reveals its correlation with the water retention characteristics and water absorption stress characteristic curve of the soil. From the above, it can be found that saturation has a significant influence on the wave velocity and Poisson's ratio of soil, and different types of soil show different variation trends with saturation. Especially for sand, it exists an optimal saturation, at which the Poisson's ratio reaches the minimum value and shows a macroscopic peak phenomenon. In the field of oil and gas seismic exploration, the Poisson's ratio of reservoir sediment is one of the important parameters to infer the gas saturation. Therefore, it is necessary to investigate the correlation between the Poisson's ratio of reservoir soil and saturation, then build a reasonable prediction model of Poisson's ratio for AVO analysis. However, there was little literature to reveal in detail the internal mechanisms of the Poisson's ratio of sand with saturation, which hinders the creation of prediction model of Poisson's ratio and detecting the gas bearing properties of sand reservoir.

Uncemented sandy sediments or sand lenses are usually good reservoirs of shallow gas in offshore environments, which can be called gas-charged sand and considered as a special type of unsaturated soil [29]. In this work, based on the theory and method of unsaturated soil mechanics and using a self-developed combined bending element–extension element–resonance column (BE-EE-RC) test system, Poisson's ratio tests under the dehumidification path were conducted on typical gas-charged sand sediments in Hangzhou Bay, China. Then, a Poisson's ratio prediction model considering the effect of saturation was developed to provide a theoretical basis for oil and gas exploration operations.

## 2. Experimental Investigations

### 2.1. Test Material

Test samples were collected from a typical shallow gas reservoir (25–30 m) in an exploration field of Hangzhou Bay. According to the ASTM standard method [30,31], the results of the particle size distribution and the specific gravity of the sample are shown in Table 1. The sample belongs to a type of silty-fine sand, and the relative density of original sand sample is 31.2%, containing a small amount (particle mass composition is 4%) of clay, with a coefficient of uniformity $C_u$ = 10, and a coefficient of curvature $C_c$ = 1.6; the particle size distribution is well-graded, which is consistent with the general characteristics of the gas reservoir formation in the Hangzhou Bay area [29].

**Table 1.** Grain size distribution of silty-fine sand.

| Specific Gravity | Particle Mass Composition (%) | | | | | $D_{50}$ (mm) | $D_{10}$ (mm) | $D_r$ |
|---|---|---|---|---|---|---|---|---|
| | >0.5 mm | 0.5~0.075 mm | <0.075 mm | <0.01 mm | <0.005 mm | | | |
| 2.68 | 2.0 | 78.7 | 19.3 | 10.0 | 4.0 | 0.243 | 0.010 | 31.2% |

### 2.2. Test Program

In order to facilitate operation and control during the test, the relative density of sand specimen is controlled to 30%, and the corresponding dry density $\rho_d$ is 1.337 g/cm³. Firstly, the soil–water characteristic curve (SWCC) of the silty-fine sand from Hangzhou bay was measured with a pressure plate extractor, and the measured data were fitted using the van Genuchten (VG) model [32], as shown in Figure 1.

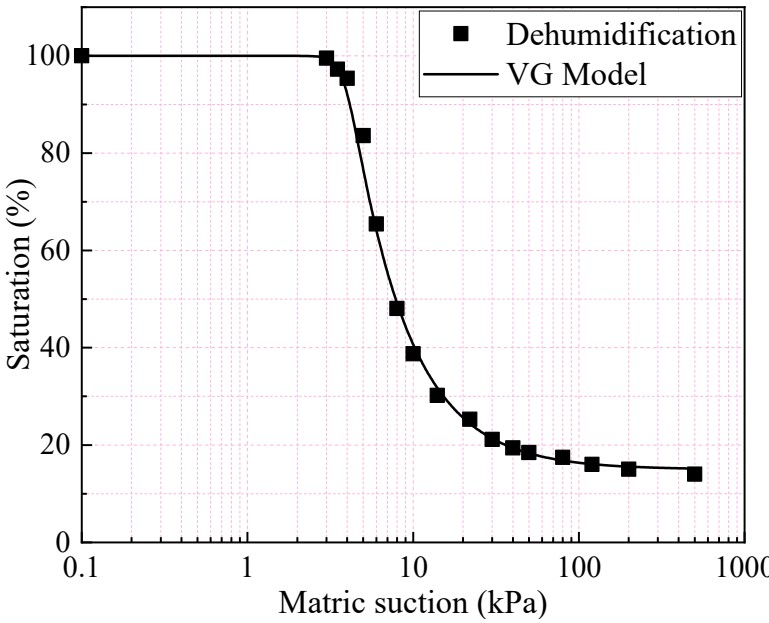

**Figure 1.** The SWCC of silty-fine sand and the prediction curve of the VG model.

The VG model can be described by Equation (3):

$$S_r = S_c + \frac{(1 - S_c)}{\left[1 + (\alpha\psi)^n\right]^m} \tag{3}$$

where $\psi$ is the matrix suction; $S_r$ is saturation; $\alpha$, $m$, and $n$ are model parameters ($\alpha = 0.2355$, $m = 0.117$, and $n = 10.839$); and $S_c$ is the residual saturation ($S_c = 15.1\%$).

According to the relationship between saturation and matrix suction in Equation (3), keeping a constant net stress, the test sand specimens with different saturations could be obtained by controlling the variation in matrix suction. Then, BE tests on sand specimens under the dehumidification path were carried out to measure the P- and S-wave velocities. The correspondence between the matrix suction and saturation of sand specimens during the test is shown in Table 2.

**Table 2.** Correspondence between the matrix suction and saturation of sand specimens.

| Matrix Suction (kPa) | 3.5 | 3.8 | 4 | 5.2 | 6 | 8 | 10 | 14 | 40 | 100 | 200 | 495 |
|---|---|---|---|---|---|---|---|---|---|---|---|---|
| Saturation (%) | 98.7 | 97.5 | 95.4 | 80 | 69 | 53 | 44 | 34 | 20 | 17 | 15 | 13 |

*2.3. Specimen Preparation and Test Method*

As shown in Figure 2, a self-developed BE-EE-RC combined test system was used to explore the effect of saturation on Poisson's ratio of silty-fine sand. The developed process, technical specifications, and calibration of the test system were introduced in detail by Yan et al. [33].

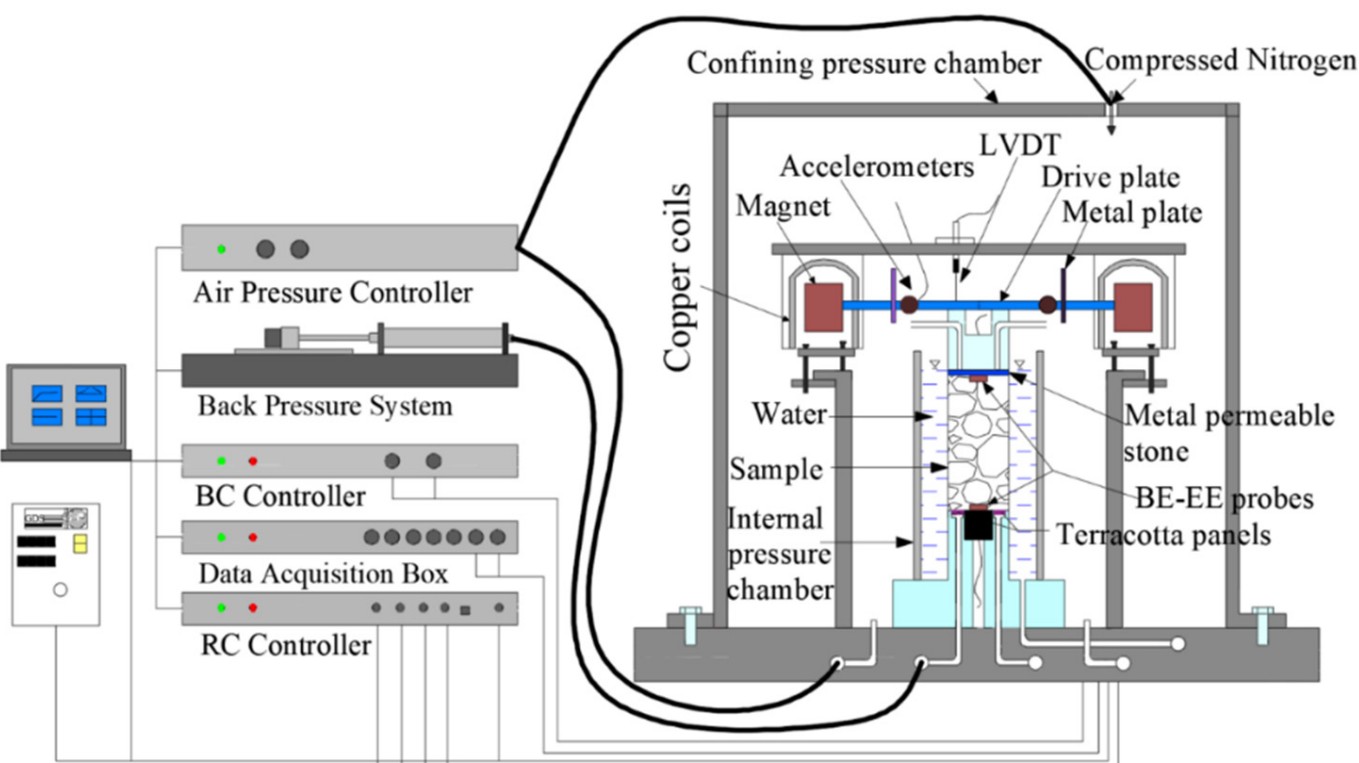

**Figure 2.** Schematic diagram of the general assembly of the BE-EE-RC test system.

In the preparation of test sand specimens ($\phi 50$ mm $\times$ 100 mm, $\rho_d = 1.337$ g/cm$^3$, $n = 0.5$), the dry density $\rho_d$ was controlled to be consistent with that used in the previous SWCC test. Firstly, sand samples with 5% water content were prepared using oven-dried sand and air-free water, and the samples were placed in a sealed container for 12 h to

make the moisture uniform. A sand specimen was formed with five layers of the sample in a saturator using the vibrating and tamping method (The tamper used to compact the material had a diameter equal to 1/2 the diameter of the mould.). To adjust the number of tamps per layer and the force per tamp until the accumulative mass of the soil placed in the mould is compacted to a known volume, each surface between two layers was carefully scarped to strengthen the connections [34]. The specimen was saturated using the vacuum saturation method in a vacuum cylinder. The saturated sand specimen was quickly frozen to reduce interference by external factors, and then installed on the pedestal of the BE-EE-RC test system. The back pressure and net stress of the specimen were maintained at 50 kPa and 100 kPa, respectively. After the specimen was thawed, based on the axis translation technique of unsaturated soil, the confining pressure was set to 150 kPa, the pore air pressure was set to 50 kPa, and the pressurization time was uniformly set to 120 min. BE-EE tests were firstly performed on the saturated sand specimen to determine the P- and S-wave velocities. After that, a series of levels of matrix suction were applied to the specimen step by step according to Table 2. The net stress and back pressure of the specimen remained unchanged, and the pore gas pressure and confining pressure changed based on the axial translation technique. When the suction equilibrium criterion was reached [35], BE-EE tests were carried out on the specimen with different saturations. Following the same method, BE-EE tests were conducted on sand specimens at net stresses of 200 kPa, 300 kPa, and 400 kPa. Based on the P- and S-wave velocities measured by the BE-EE tests, the value of Poisson's ratio can be obtained using Equation (2).

*2.4. Measurement of Arrival Time*

During the BE-EE tests, the P- and S-wave velocities were determined by the measured upper and lower BE tip distance *d* and the propagation time *t*, as shown in Equation (4):

$$V_S = \frac{d}{t_S}, V_P = \frac{d}{t_P} \tag{4}$$

As the near-field effect (the frequency dependence of the S-wave motion due to the P-wave interference) influences the received signal, correctly assessing the arrival time of the S-wave is difficult [36,37]. In order to eliminate the near-field effect, BE tests were performed at five different input frequencies: 5 kHz, 10 kHz, 12.5 kHz, 20 kHz, and 25 kHz, as shown in Figure 3. It was found that the near-field effect was minimized by increasing the frequency of the input signal; when the excitation frequency is greater than 20 kHz, the shear waveform is basically stable, and the effect of maintaining the increase in the excitation frequency to minimize the near-field effect is no longer obvious.

In addition, compared to the BE test, the reliability of the RC test in investigating the small strain properties of soils has been widely accepted [38]. Therefore, the RC test was used to calibrate the BE test to eliminate the drawback of artificial uncertainty in determining the arrival time of the S-wave. In order to avoid disturbance of the specimen by the RC test, the BE test was carried out first, and the RC test was carried out after the BE test. The RC test was used to calibrate the BE test to eliminate the drawback of artificial uncertainty in determining the arrival time of the S-wave. Figure 3 shows the comparison results between the RC test and BE test at different saturations and net stresses. The result indicates that the arrival time of the S-wave selected according to the start–start (S–S) method [39] in the BE test was in the best agreement with the inverse analysis result of the RC test, and the results of the RC test and BE test were basically consistent when the excitation frequency was 12.5 kHz. Ultimately, the excitation frequency of the BE test was selected as 12.5 kHz in this work, and the S-wave arrival time was determined using the S–S method.

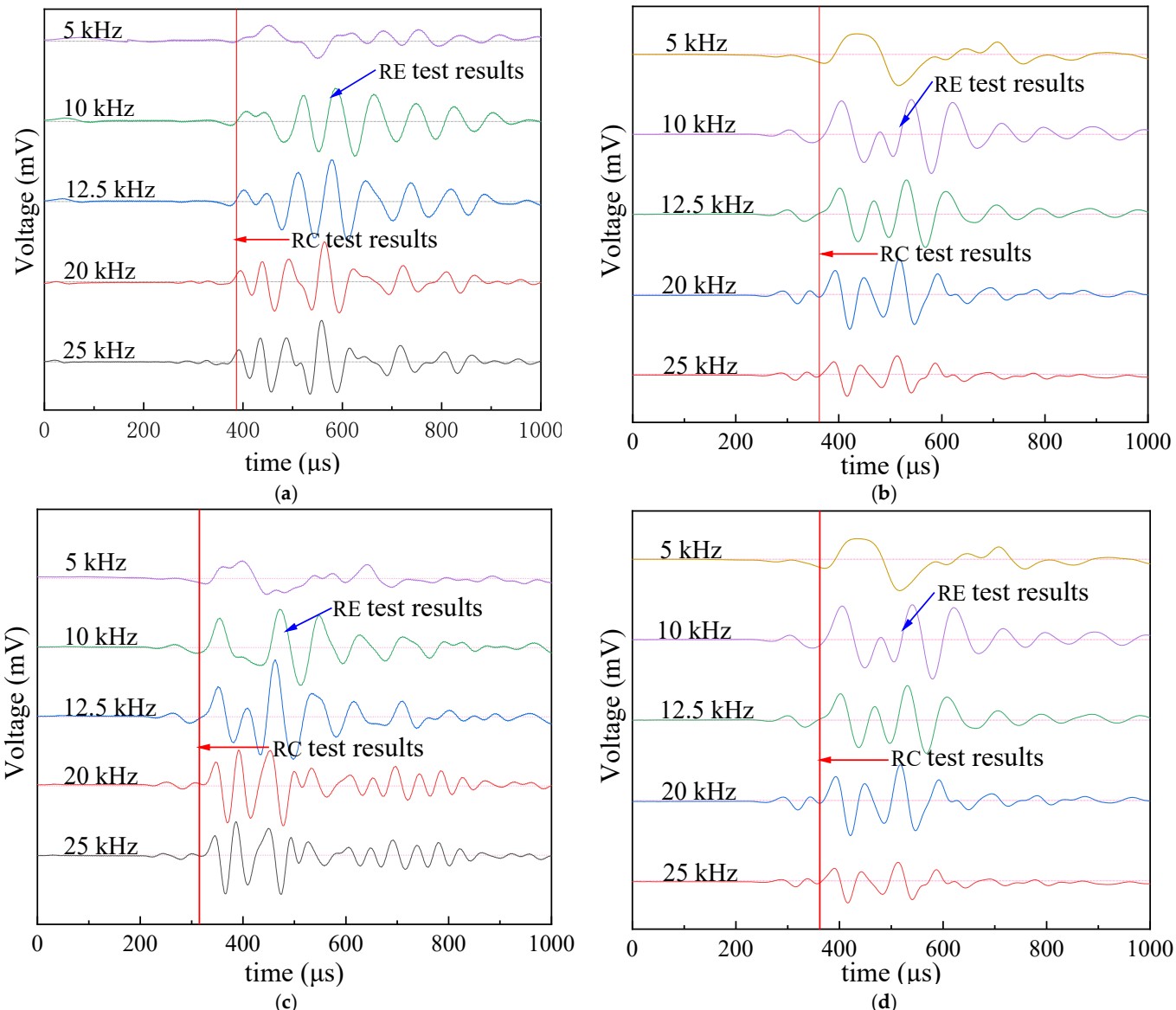

**Figure 3.** The results of the BE-RC test of silty fine sand under different net stresses. (**a**): Sr = 0, net stress = 200 kPa; (**b**): Sr = 1, net stress = 200 kPa; (**c**): Sr = 0, net stress = 400 kPa; (**d**): Sr = 1, net stress = 400 kPa.

### 3. Test Results

#### 3.1. P-Wave Velocity and S-Wave Velocity at Different Saturations

Figure 4 shows the variation curves of the S-wave with saturation for the Hangzhou Bay silty-fine sand, where the scatter points are the measured values (The input frequency of S-wave was 12.5 kHz.). The results indicate that the variation in $V_s \sim S_r$ curves under different net stresses is basically similar. With the decrease in saturation, $V_s$ is in a stable state when the saturation is between 100% and 95%; in the range of 95%~80%, the S-wave velocity starts to increase slowly, and the change is not significant. When the saturation continues to decrease, the S-wave velocity gradually increases and reaches a maximum at the optimal saturation $S_{r(opt)}$; the S-wave velocity decreases rapidly when the saturation is less than the optimal saturation. At the same saturation, the higher the net stress, the higher the S-wave velocity.

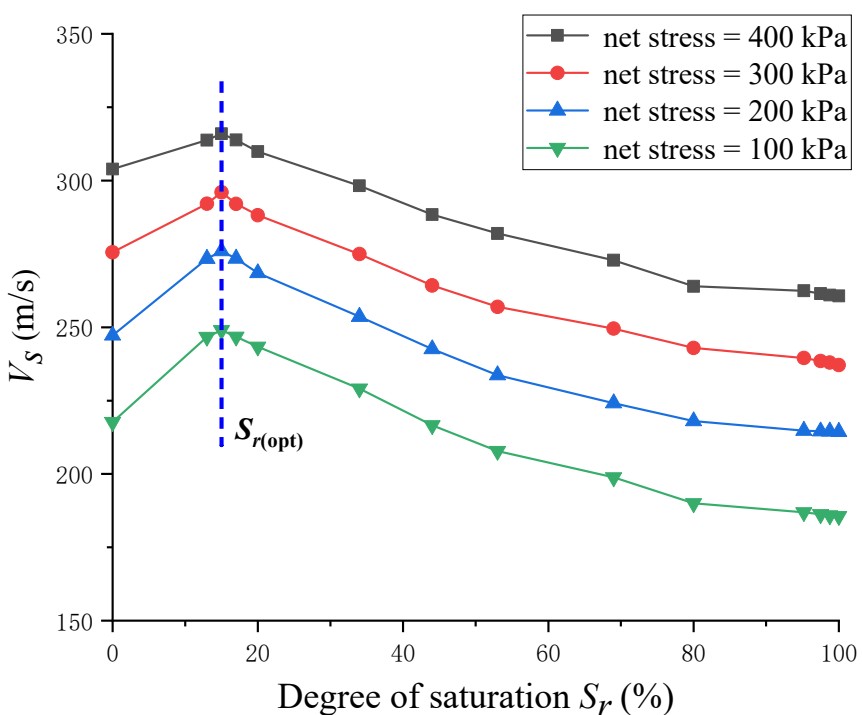

**Figure 4.** Variation in the S-wave velocity with the degree of saturation at different net stresses.

Figure 5 shows the P-wave variation curves with saturation for the silty-fine sand, including the measured and theoretical predicted values (The input frequency of P-wave was 20 kHz). Soil can be regarded as a porous medium. If no relative movement occurs between the pore fluid and solid particles, the elastic wave velocities can be theoretically expressed as a function of the effective elastic moduli of soil at a given saturation degree [21]. For the P-wave and S-wave, the velocity $V_p$ and $V_s$ can be described in terms of the effective bulk modulus of the soil $K$, effective shear modulus of the soil $G$, and density of the soil $\rho$, as shown in Equations (5) and (6); while $K$ is determined using Equation (7) [40]:

$$V_P = \sqrt{\frac{K + \frac{4}{3}G}{\rho}} \tag{5}$$

$$V_S = \sqrt{\frac{G}{\rho}} \tag{6}$$

$$K = K_{sk} + (1 - \frac{K_{sk}}{K_s})^2 / (\frac{n}{K_f} + \frac{1-n}{K_s} - \frac{K_{sk}}{K_s^2}) \tag{7}$$

where $K_{sk}$ is the effective bulk modulus of the soil skeleton, $K_s$ is the bulk modulus of solid grains, $K_f$ is the bulk modulus of the pore fluid, and $n$ is porosity. $K_{sk}$ is determined by measuring the $V_s$ and $V_p$ values of dry soil. For a given value of $S_r$, $K_f$ can be determined using Equation (8):

$$K_f = (\frac{S_r}{K_w} + \frac{1-S_r}{K_a})^{-1} \tag{8}$$

where $K_w$ is the bulk modulus of water, and $K_a$ is the bulk modulus of gas.

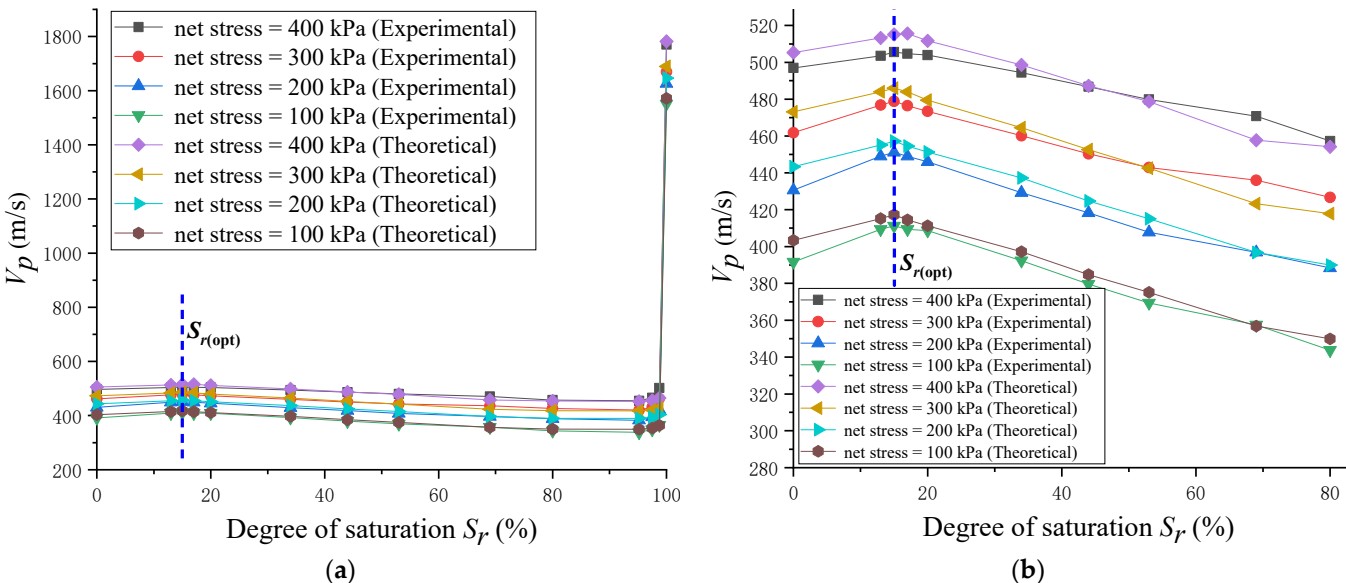

**Figure 5.** Variation in the P-wave velocity with the degree of saturation at different net stresses. (**a**): Saturation range is 0~100%; (**b**): Saturation range is 0~80%.

According to Equation (8), $K_f$ can be determined based on $K_w$, $K_a$, and $S_r$. Then, $K_f$ is substituted into Equation (7) to obtain the $K$ value. Based on the experimentally measured value of $V_s$, $G$ can be obtained using Equation (6). Then, the values of $K$ and $G$ can be substituted into Equation (5) to obtain the value of $V_p$ associated with the given values of $S_r$ and $n$. In this study, the values of $K_w$ and $K_a$ were taken as 2.18 GPa and 142 kPa, respectively [41]. Because the sand particles are primarily composed of quartz mineral, the value of $K_s$ was taken as 36.6 GPa [42].

The results show that the theoretical predictions based on Equations (5)–(8) with respect to the variation in $V_p$ with $S_r$ match well with the observed experimental data. The theoretical predictions verified the reliability of test results. At high saturation (gas content 0–2%), the P-wave velocity varies sharply and generates a sudden drop; as the saturation continues to decrease, the wave velocity decreases slowly and gradually stabilizes. When the saturation continues to decrease, the P-wave velocity gradually increases and reaches a peak at the optimum saturation $S_{r(opt)}$.

### 3.2. Poisson's Ratio ν and Prediction Model

#### 3.2.1. Influence of Saturation on Poisson's Ratio

Figure 6 shows the curve of Poisson's ratio with saturation. According to the results, it is found that the variation law of Poisson's ratio with saturation is basically similar under the different net stresses. Beginning from the saturated state, as the saturation decreases, gas invades the pore space, and the soil transitions to the unsaturated state. Poisson's ratio shows a sudden drop at first, and gradually tends to a stable value; with the continuing decrease in saturation, Poisson's ratio starts to decrease and reaches the minimum value at the optimum saturation ($S_{r(opt)}$ is 15%). Then, Poisson's ratio gradually increases when the saturation is lower than the optimal saturation. Figure 7 shows the variation curves of Poisson's ratio with the net stress at different saturations. In the range of the test net stresses (100 kPa, 200 kPa, 300 kPa, 400 kPa), Poisson's ratio shows a linear relationship with the net stress, which slightly decreases with the increase in net stress. The result indicates that the influence of the stress level on Poisson's ratio is smaller than that of saturation.

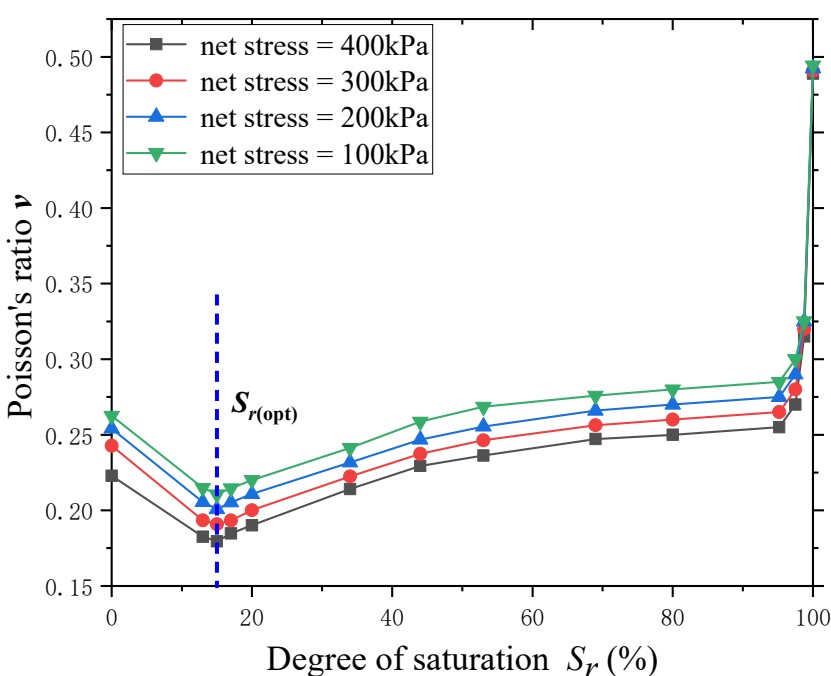

**Figure 6.** Poisson's ratio variation curve with saturation.

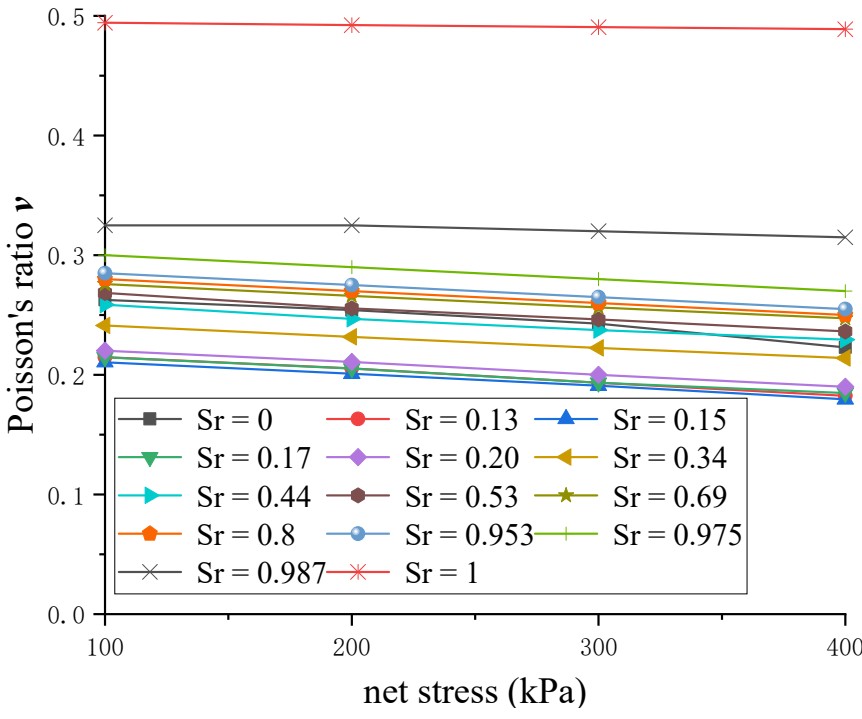

**Figure 7.** Poisson's ratio variation curve with net stress.

### 3.2.2. Influence Mechanism of Saturation on Poisson's Ratio

As is well known, the matrix suction is a function of saturation in soil–water characteristic curve (SWCC), and they are closely related to each other. The SWCC is an important property to study unsaturated soils, which reflects much more information including the soil structure, soil particle composition, pore size, and pore distribution. The mechanical properties of unsaturated soil are influenced by SWCC [43,44]. As can be seen in Figure 6, Poisson's ratio of the silty-fine sand is closely related to saturation; thus, it also has a strong relationship with SWCC. The SWCC of soil can be divided into four different desaturation stages based on internal mechanisms from capillary action and sorption in the particles,

namely the boundary effect stage I, the primary transition stage II, the secondary transition stage III, and the residual saturation stage IV [45]. As shown in Figure 8, stage I is determined mainly based on the air-entry value R1 of the soil, and the dividing point between stage II and stage III is the inflection point R2 in the SWCC. The water-holding characteristics of the soil are mainly controlled by the pore size of the soil, so the pore distribution is often used to explain the macroscopic water-holding characteristics [46,47]. According to the variation in the pore distribution density of the pore size, the maximum value of its pore distribution density often corresponds to the inflection point R2 in SWCC; stage III and stage IV are determined mainly based on the residual saturation R3 in SWCC. The different distribution patterns of the pore water within the four stages are shown in Figure 9.

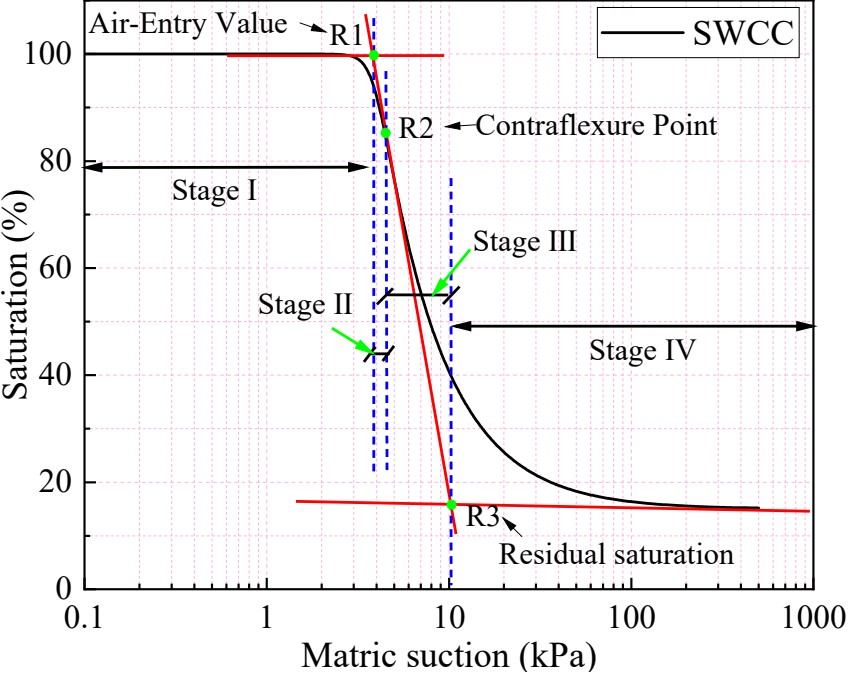

**Figure 8.** Matrix suction versus degree of saturation.

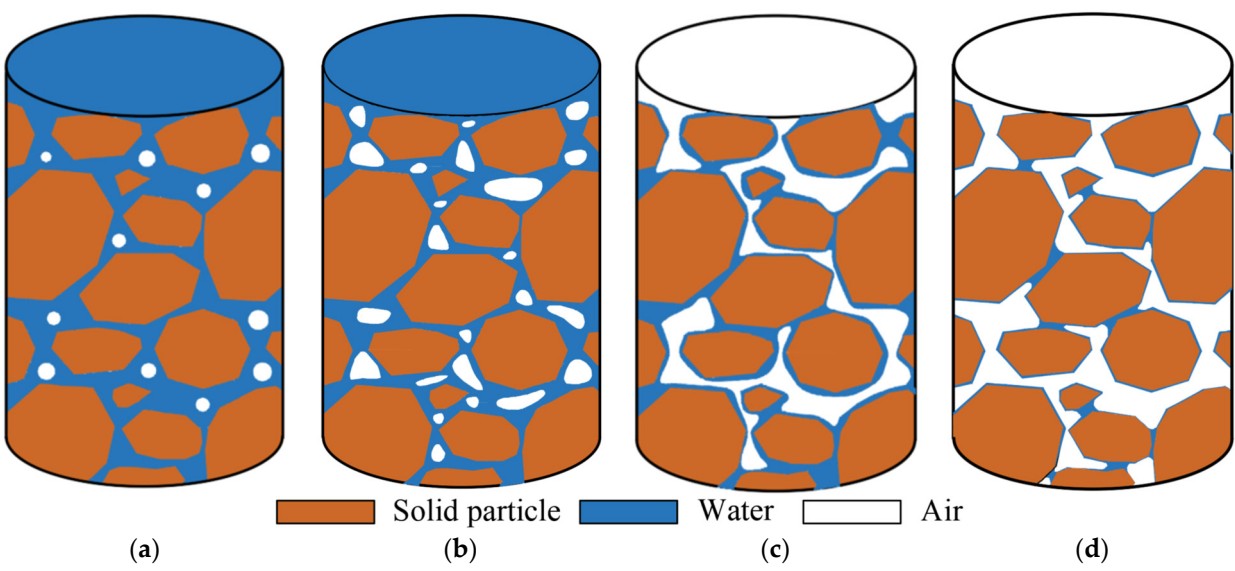

**Figure 9.** Distribution pattern of pore water within the four stages. (**a**): Stage I; (**b**): Stage II; (**c**): Stage III; (**d**): Stage IV.

In stage I ($S_r$ is 100~95%), the whole pores in the sand particles are filled with water and interconnected. The pore water distribution pattern is shown in Figure 9a. Beginning from the saturated state, with the decrease in saturation, a small amount of gas first enters into the large pores in the sand and forms several small bubbles, which can flow with the pore water. During stage II ($S_r$ is 95~80%), the internal water distribution pattern is shown in Figure 9b. As the saturation continues to decrease, the volumes of closed bubbles increase, resulting in the pore water in the larger pores being displaced gradually, and the closed smaller bubbles beginning to enter the small pores. In stage III ($S_r$ is 80~15%), the internal water distribution pattern is shown in Figure 9c. The gas bubbles become larger and larger, the water–gas interface begins to come into contact with the sand particles, and the contact angle starts to form. At this time, both the water phase and gas phase in the pores of sand are connected with each other. The internal water distribution pattern is shown in Figure 9d. When the soil saturation falls below the optimum saturation, it is in the stage IV ($S_r$ is 15~0%). The pore gravity free water almost disappears under the action of matrix suction and tension suction, being in the form of annular water around the particles. This is called the water-closed state. The areas of annular water are independent of each other, but still in contact at the particle contact corner. With further reduction in the free annular water, the water phase only remains the bonded water wrapped around the particle surface. Because of the small specific surface area of sand particles, the surface adsorption charge is lower; thus, the adsorption water layer is thinner, meaning that the weak adsorption water layer also disappears quickly.

According to the different stage of SWCC, the $v$~$S_r$ variation curve can also be divided into four similar stages, as shown in Figure 10. According to the different distribution patterns of the pore water within the four stages, the mechanism of Poisson's ratio variation with saturation is analyzed from the interparticle stresses.

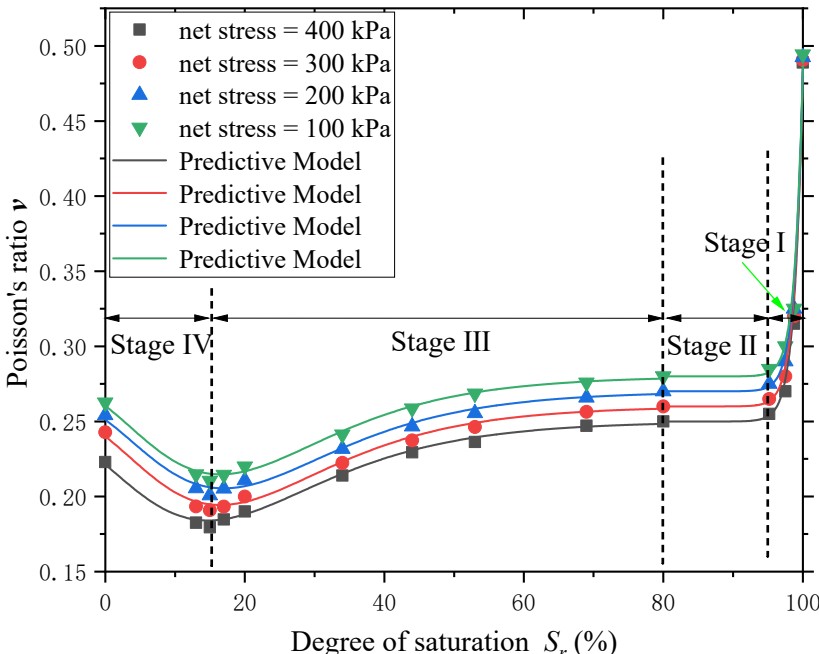

**Figure 10.** Poisson's ratio variation curve with saturation and predictive model.

In stage I, the bubbles in the pores are not in contact with sand particles; thus, the surface tension is transmitted on the surface of pore water and balanced with pore water pressure, which is not a direct acting force on sand particles. Therefore, because there is no significant contribution to the interparticle stress [33], the resistance of soil skeleton stays constant. The main reason for the sudden drop in Poisson's ratio at this stage is that the pore fluid modulus $K_f$ has an obvious reduction due to the entry of gas bubbles. Because the small strain bulk modulus $K$ of soil is mainly composed of three parts: sand particle

modulus $K_s$, skeleton modulus $K_{sk}$, and pore fluid modulus $K_f$ (fluid modulus of water and gas $K_w$, $K_a$), as shown in Equation (7), $K$ also generates a sudden drop with the entry of gas bubbles. According to the definition of bulk modulus and Poisson's ratio, under the same stress, as the bulk modulus decreases, the volume contraction increases, the ability of the sand specimen to resist compression weakens, and the deformation ability increases. Therefore, once gas enters, Poisson's ratio ν first decreases sharply.

In stage II, the interparticle forces in the sand are the same as in stage I. The change in the bulk modulus due to gas entry is stabilized. The matrix suction is smaller, and it is not affected by other forces. Thus, Poisson's ratio in this stage basically maintains a stable value.

In stage III, the change in the bulk modulus caused by gas intrusion is small and almost negligible due to the entry of a large amount of gas. As the gas phases in pores of the specimen begin to connect with each other, the pressure of gas or water phase can be transferred, independently. Surface tension is transferred and acts on sand particles through the curved liquid surface around different particles, and it decreases with the decreasing radius of the curved liquid surface. However, due to the increase in matrix suction, the combined suction force gradually increases, which contributes to the increase in intergranular stress. Thus, the resistance of sand to lateral deformation gradually increases, resulting in the decrease in Poisson's ratio. When the water phase around the particle contact starts to form in the water-closed state, the saturation is located at the optimum saturation $S_{r(opt)}$, and the combined suction reaches a peak value, while the ability of sand skeleton to resist deformation reaches the maximum. Therefore, Poisson's ratio of the sand specimen shows the minimum value.

In stage IV, although the distance between particles tends to be minimal, the interparticle spacing of sand is still too large and beyond the sphere of influence of short-range forces [33]. Thus, only the tensional suction force acts on particles. Because of the small curvature radius, the combined suction force between the particles generates a sudden drop, leading to a rapid decrease in intergranular stress. The ability of the sand skeleton to resist deformation weakens, so that the Poisson's ratio gradually increases.

### 3.2.3. Prediction Model of Poisson's Ratio with Saturation

According to the effect of saturation on Poisson's ratio and the intergranular stress variation, the $v{\sim}S_r$ variation curve of silty-fine sand can be divided into two parts, where the demarcation point is the inflection point R2 in the SWCC. One part includes stage I and stage II, and the intergranular stress is mainly influenced by matrix suction. The other part includes stage III and stage IV, and the intergranular stress is influenced by both matrix suction and surface tension. The coordinates of the inflection point R2 ($\psi_i$, $S_i$) can be obtained using Equation (9) [48], based on Equation (3):

$$\begin{cases} \psi_i = \frac{1}{am^{1/n}} \\ S_i = \frac{1}{(1+1/m)^m} \end{cases} \tag{9}$$

Figure 10 shows the Poisson's ratio variation curve with saturation, where the scattered points are the measured values of the test, and the solid line is the prediction model of Poisson's ratio of silty-fine sand. It shows that the prediction model can sufficiently reflect the variation in Poisson's ratio with saturation. Poisson's ratio is significantly correlated with saturation, and ignoring the influences of saturation on Poisson's ratio may lead to serious identification errors in oil and gas industry exploration. The prediction model is based on the variation law of Poisson's ratio, with saturation measured using the curve-fitting procedure; Poisson's ratio of silty-fine sand at any saturation can be determined using Equations (10) and (11):

$$\nu = \begin{cases} A \cdot e^{(-100 \times S_r/B)} + \nu_{inp} & S_r \geq S_{r(inp)} \\ \nu_{inp} + C \cdot e^{(-e^{(-D)}-D+1)} & S_r < S_{r(inp)} \end{cases} \tag{10}$$

$$D = 100 \times (S_r - S_{r(opt)})/E \tag{11}$$

where $A$, $B$, $C$, and $E$ are fitting parameters, $S_{r(inp)}$ is the degree of saturation corresponding to the inflection point R2 in the SWCC; $v_{inp}$ is Poisson's ratio corresponding to this saturation $S_{r(inp)}$; and $S_{r(opt)}$ is the optimal saturation, which can be obtained using Equation (12) [22]:

$$S_{r(opt)} = 0.01 \times \left( -6.5 \log_{10}^{D_{10}} + 1.5 \right) \tag{12}$$

where $D_{10}$ is the particle size such that 10% of the soil by mass is finer than that size. The detailed fitting parameters are shown in Table 3.

**Table 3.** Fitting parameters of the predictive model.

| Parameters | σ = 400 kPa | σ = 300 kPa | σ = 200 kPa | σ = 100 kPa |
|---|---|---|---|---|
| $A$ | $4.5165 \times 10^{-38}$ | $2.0485 \times 10^{-38}$ | $8.6409 \times 10^{-39}$ | $4.0231 \times 10^{-40}$ |
| $B$ | −1.183 | −1.173 | −1.161 | −1.122 |
| $C$ | −0.06593 | −0.06564 | −0.06459 | −0.06524 |
| $E$ | 13.871 | 13.408 | 13.614 | 13.354 |
| $v_{inp}$ | 0.252 | 0.261 | 0.272 | 0.281 |
| $R^2(S_r \geq S_{r(inp)})$ | 0.9911 | 0.9898 | 0.9887 | 0.9818 |
| $R^2(S_r < S_{r(inp)})$ | 0.9839 | 0.9798 | 0.9779 | 0.9828 |

### 3.2.4. Validation of the Prediction Model

In order to verify the applicability and reliability of the prediction model proposed in this work, based on the SWCC curve and the measured P- and S-wave velocities of Edosaki sand specimens at different saturations [49], the saturation corresponding to the coefficient of determination (R2) of the model is 78%, and the Poisson's ratio values at different saturations were obtained using Equation (2), as shown in Figure 11. The scatter points are the measured values, and the solid line is the model-predicted value. According to the fitting results of the prediction model, it can be seen that Poisson's ratio under different net stresses with the variation law of saturation has a good fit, and the overall prediction trend appears well, which verifies the reliability of prediction model.

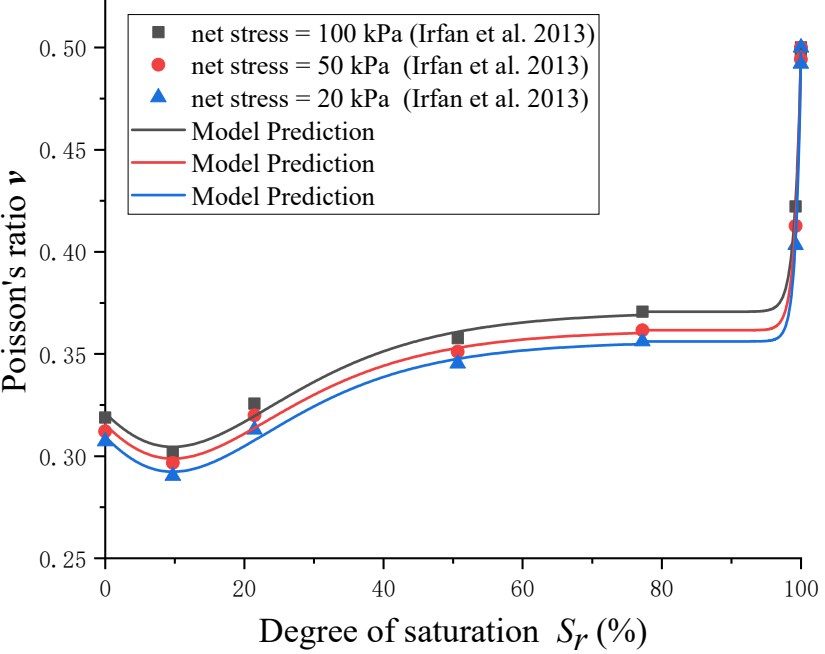

**Figure 11.** Variation curve and model prediction of Poisson's ratio for Edozaki sand [49].

## 4. Conclusions

To investigate the effect of saturation on Poisson's ratio of reservoir sediments is of important engineering significance for the detection of gas-bearing reservoirs. Based on the theory and method of unsaturated soil mechanics, the Poisson's ratio evolution law with saturation for reservoir sandy sediments in Hangzhou Bay was investigated, and the internal mechanisms causing the variation in the interactions among the sand particles, pore water, and gas were revealed. Then, a prediction model for Poisson's ratio with saturation was proposed for silty-fine sand. The main conclusions are as follows:

(1) At the same degree of saturation, the higher the net stress, the higher the P- and S-wave velocities. Saturation has a significant effect on both a $V_S$ and $V_P$; in the high saturation range, saturation is more sensitive for $S_{r(opt)}$ than $V_S$, and $V_P$ generates an abrupt decrease; with the decreasing saturation, both $V_S$ and $V_P$ gradually increase and reache a peak value at the optimum saturation.

(2) The curve of Poisson's ratio versus saturation can be divided into four stages: in the boundary effect stage I, Poisson's ratio generates a sudden drop with a small decrease in saturation; in the primary transition stage II, the variation in saturation has a small effect on Poisson's ratio; in the secondary transition stage III, the matrix suction and tension suction play a major role, and Poisson's ratio decreases with decreasing saturation, reaching a minimum at the optimum saturation; in the residual saturation stage IV, the combined suction decreases rapidly, and Poisson's ratio increases with the reduction in saturation.

(3) A prediction model for Poisson's ratio of silty-fine sand considering saturation variation was proposed, which can provide a theoretical basis for reasonably inferring the shallow gas-bearing state in the Hangzhou Bay.

**Author Contributions:** Conceptualization, K.Y. and Y.W. (Yong Wang); Funding acquisition, Y.W. (Yong Wang); Investigation, K.Y., X.L., and Z.Y.; Methodology, K.Y.; Resources, Y.W. (Yong Wang), X.L. and Y.W. (Yanli Wang); Validation, X.L. and Z.Y.; Writing—original draft, K.Y.; Writing—review and editing, Y.W. and Y.W. (Yanli Wang). All authors have read and agreed to the published version of the manuscript.

**Funding:** This research was funded by the Natural Science Foundation of China, grant numbers 51979269 and 52127815; Wuhan Research Program of Application Foundation And Frontier Project, grant number 2020010601012181.

**Institutional Review Board Statement:** Not applicable.

**Informed Consent Statement:** Not applicable.

**Data Availability Statement:** The data presented in this study are available on request from the corresponding author.

**Conflicts of Interest:** The authors declare that they have no conflict of interest.

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
