# Peer review of "Experimental Study on Poisson’s Ratio of Silty-Fine Sand with Saturation"

_jmse, doi:10.3390/jmse11020427_

Round 1
Reviewer 1 Report
This work presents laboratory and modelling data to predict the evolution of Poisson's ratio with degree of saturation for unconsolidated sands. Authors used samples derived from the Hangzhou Bay to build a model able to predict the Poisson's ratio of sands. Overall, the manuscript is clear in its aims and it well presents the experimental and the modelling procedure. However, I feel that it need some minor revisions before publications. Below the authors can find more specific comments which, I hope, can help them in improving the manuscript.
General Comments:
?ν?1. I feel that the abstract misses a concluding paragraph summarising the general implications of their work. I would suggest adding it for completeness.
2. In places, in the "Introduction" section the authors miss to fully explain terms then used in the rest of their work. This makes hard to read this section. Examples of this are fully explained in the "Detailed comments" below.
3. In general, figures' captions are not clear and do not fully explain the related figures. I would suggest the authors to rewrite them in a way that the figure and the caption are self-explanatory.
Detailed Comments:
Line 13: what do you mean by "self-explanatory"? Do you mean that the experimental set-up is made in-house?
Line 19: the ? symbol is used here but no reference to its meaning are given. Please explain it.
Line 38: here the authors say that Poisson's ratio is "more conducive", but we do not know the term of comparison for it. Can you please indicate compared to which other parameter the Poisson's ratio has better use?
Lines 51-52: the sentence here is very vague and therefore not very clear. Please fully explain which are these "different ways" you are referring to, and also explicitly state the range of strain.
Lines 55-57: This sentence is not very well-written, thus resulting not clear. What do you mean by "easily disturbing"?
Line 58: I do not think that "prone" is the right word choice. Do you mean that there is a tendency of using Poisson's ratio derived from seismic data?
Line 65: Please fully explain the meaning of "optimum degree of saturation". As this is used throughout the manuscript, it very important that the reads have a clear definition of this parameter. How is it defined?
Lines 85-87: The sentence here is not very clear. which are these "macroscopic phenomena" you are referring to? please explain fully. Why there is not reason for explaining the internal mechanisms? Also, please cite the appropriate literature for interested readers.
Line 102: Please explain what ASTM stands for?
Line 104: here you are referring to a "small amount", please report how much. As written it is very vague.
Line 105: Please further explain the meaning of the two coefficients used here (Cu and CL). How are these calculated?
Line 106: I would suggest writing "well-graded" and not 'graded well'
Line 128: As for the abstract, what do you mean by "self-developed"? This will be repeated in the conclusion.
Line167: This figure is not clear. Can you please indicate which graph corresponds to BE and which to RC tests?
Line 171: please add a reference at the end of the sentence.
Lines 171-172: this sentence is not clear.
Line 184: Here you referring to the S-wave velocity results. Can you please state at which input frequency? Similarly for Figure 4, please state the input frequency.
Line 193: in Figure 4 can you please explain in the text why is it important to use the Sr(opt) as threshold?
Line 257: Reference needed at the end of the sentence here.
Line 264: Figure 9 needs a legend explaining the colours.
Reviewer 2 Report
This paper aims to investigate the Poisson's ratio variation with respect to saturation ratio of a silty‑fine sand sample taken from Hangzhou Bay, China.
My comments about the study are as follows:
· It would be better for the manuscript to be proofread by a native English speaker.
· At the end of Introduction section the reason for this study should be explained more powerful with supporting arguments. Especially insisting on the filling the absent points in the literature. Also enhancing the introduction section would be better.
· Line 104: The clay content is not a small amount (@ 20%). For this clay fraction Cu and Cc parameters are not significant. For the classification of such a sample consistency limits results should be used.
· Line 113: How could the Authors obtain the dry density of sample as 1.337 g/cm3? Explanation of the methodology would be better.
· Line 116: In the figure caption the long version of VG Model is more appropriate.
· Line 119: In the manuscript the saturation ratio values are presented as percentages. In order to be more harmonic presenting the Sc value should also be in percentages.
· Line 120: the “Eq1” phrase should be corrected as “Eq3”.
· Line 132: The size of the figure and the font sizes should be enlarged.
· Line 134: Which energy is used for the preparation of the samples to achieve the target dry density? The sample preparation method should explained in more detail. Did the Authors check the final saturation ratios of the samples as duplicate control specimens? Also are there any duplicate tests performed especially for extreme test results?
· Line 161: using the phrase “sometimes” is not suitable in such a paper. Those exceptional situations should be explained.
· Line 195: In figure 5 there are two sub figures even the second one is a section of first graph. In such figures using A and B as sub figures and their explanation in the captions would be better.
· Line 214 and 215: The selection of the parameters in these sentences should be explained in detail.
· Line 262: Figure 8 is almost same with Figure 1. If this figure is used for the definitions of some parameters and stages of the saturation variation then this figure should be redrawn in a more a representative sketch.
· Line 267, 271, 274 and 278: The figure referencing is wrong. Probably they should be Figure 9.
· Line 292: Figure 10 is almost same with Figure 6. Only stage boundaries are presented as a difference with Figure 6.
· Line 353 and 354: What is the reason for the usage of two different Tables? Table 3 and Table 4 should merged.
· Line 364: If the data points in figure 11 is taken from another references, they should be mentioned in the figure. Presentation of determination coefficients (R2) of the models might be better to support the power of prediction models.
· Conclusion section should be enhanced.
· There is duplicate entries for same reference (Ref 10 and Ref 18). References section should be checked.
